# Corporate Sustainability: The Impact of Environmental, Social, and Governance Performance on Corporate Development and Innovation

Defang Ma [1], Liangwei Li [1,*], Yuxi Song [2], Mengkai Wang [3] and Qiaowen Han [1]

[1] School of Management, Capital Normal University, Beijing 100048, China; 6346@cnu.edu.cn (D.M.); 15254105956@163.com (Q.H.)

[2] School of Economics and Management, Henan University of Technology, Zhengzhou 450007, China; songyuxi1008@163.com

[3] School of Business, Renmin University of China, Beijing 100872, China; mengkaiw@ruc.edu.cn

* Correspondence: 2222902013@cnu.edu.cn

**Abstract:** As a comprehensive concept that integrates the environment, society, and corporate governance, little is known about whether and how Esg affects firm development, as the concept of sustainable development is deepened and promoted. Therefore, the purpose of this paper is to investigate the impact of Esg performance on corporate development. This paper selects the data of A-share-listed companies from 2010 to 2020 as samples, utilizes the linear regression model to empirically study the impact mechanism of Esg performance on enterprise development, and considers transmission pathways. It is found that the development of high-technology firms is more significantly affected by Esg performance than the development of non-high-technology firms. It is further found that Esg performance can promote enterprise development by reducing financing constraints. Meanwhile, corporate innovation can enhance the promotion effect of Esg performance on corporate development. After the robustness tests of instrumental variables and the lagged effects, the research conclusions still hold.

**Keywords:** ecology; society; governance

## 1. Introduction

There is a close link between environmental, social, and governance (Esg) performance and corporate innovation and development [1,2]. Esg performance has gradually become a mainstream topic in the financial and business worlds [3], with more and more investors attaching importance to corporate performance in relation to Esg performance. Specifically, stakeholder-based corporate decision-making theory suggests that a high level of Esg performance makes it easier for corporations to be recognized and regulated by stakeholders and that the altruistic signals conveyed can attract more investment and financing opportunities [3–5], thus enhancing corporate innovation [6]. In addition, good Esg performance and active fulfillment of social responsibility reflect a change in corporate governance thinking, not only in terms of the quality of internal governance but also in terms of meeting the information needs of both sides of the market. This is of great significance to both corporate and individual decision-making [7], particularly in realizing cooperation and sharing [8]. This idea can drive innovative thinking and stimulate the potential for cooperation and innovation, thus promoting the efficiency of corporate innovation. Innovation is the inherent driving force of economic development. As a macroeconomic micro-body, enterprise innovation helps to promote the upgrading of industrial structures to optimize the enterprise value creation mode [9], stimulate consumer demand for diversified products, and achieve a win-win situation in the market. In the era of the digital economy, enterprise innovation makes the boundaries of various industries gradually blur [10,11],

accelerates the integration of knowledge and resource elements of different industries, and gradually stimulates economic vitality, causing the dynamic evolution of development spillover effects and thus promoting development [12].

At present, under the guidance of the concept of sustainable development [13,14], the innovation process of enterprises, whether based on technological innovation or market-oriented business model innovation, pays more attention to the values and the expectations of enterprise stakeholders and emphasizes more of the creation of economic, social, and environmental responsibilities and other diversified and comprehensive values based on innovation, as well as on the provision of effective tools and resource bases based on enterprise innovation for the enhancement of market performance and competitiveness. The development of enterprise innovation reflects the speed and the quality of achieving enterprise innovation [15]. The development reflects the speed and the quality of realizing enterprise innovation; therefore, innovation is an important strategic decision for enterprises; vigorously promoting innovation through capital investment is the key for enterprises to maintain competitive advantage and enhance enterprise value [16,17].

Existing studies have found that companies with good Esg performance tend to have good performance in terms of financial, operational, and social responsibility [18]. This is because these companies pay more and more attention to economic, social, and environmental issues as they grow to a certain scale and because they have a certain degree of foresight to make timely predictions and prevent possible risks in the future. In order to maintain their long-term competitive advantage, these enterprises will put quality improvement and enterprise innovation in an important position when their productivity reaches a certain level. In addition, they are concerned about environmental and social issues, and they convey a positive image to society. In view of this, this paper empirically analyzes the governance effect of Esg performance based on the perspective of corporate development, and it further examines the specific channels of influence as well as the differences in the role of Esg performance in different contexts.

This study can mainly help enterprises reduce financing constraints, and it can provide financial support for innovative activities [18]. It fully utilizes the knowledge-sharing mechanism to provide technical support for member enterprises' enterprise innovation activities. Moreover, it is conducive to the realization of resource sharing and synergy effects, which can promote enterprise innovation. It achieves the maximization of economic, environmental, and social benefits so as to better promote development.

In order to address these issues, this study presents its research hypotheses through theoretical analysis in the second part. In the third part, China's A-share listed companies from 2010 to 2020 are taken as the research samples, and the selection of variables is determined. The fourth part then adopts empirical analysis to systematically test the impact of Esg performance on corporate development and to explore the possible paths of corporate development through Esg performance. The fifth part offers a comparative analysis with other scholars' studies. The sixth part is the discussion, and the seventh part concludes the paper.

## 2. Theoretical Analysis and Research Hypothesis

According to stakeholder theory, enterprises should not only focus on shareholders but also on the demands of various stakeholders, such as employees, customers, suppliers, and the public. The Esg performance of enterprises involves the dimensions of the environment, society, and governance, which are precisely the concerns and considerations of stakeholders. Therefore, by paying attention to Esg performance, enterprises can, to a certain extent, satisfy the needs of these stakeholders, which can lead to a better reputation, image, and financial support from long-term investors, thus promoting the long-term stable growth and development of enterprises.

According to the theory of sustainable development, the environmental performance, the social responsibility performance, and the corporate governance performance that comprise Esg are highly overlapping, with the three factors of society, environment, and

economy emphasized by the theory of sustainable development; the enterprise itself is a manifestation of following the principle of sustainable development while realizing Esg behavior [19]. It means that the sustainable concepts of society, environment, and governance are deeply integrated with business processes and with value-creating activities and that the results are not only reflected in corporate products but also in corporate culture, the organizational structure, and the efficiency of the allocation of various factors that can be further improved [18]. Therefore, enterprises actively practicing the Esg development concepts are not only performing well in non-financial areas but their practice is also a positive signal that they will achieve a high level of sustainable development.

Existing studies have found that enterprises with good Esg performance show a higher tendency to disclose information and are willing to convey more open and transparent information to the external market. Nowadays, information resources have become one of the most important elements of the new-generation resource system [5,20], and resources naturally flow to high-resource position modules as the added value of enterprises' products increases. In addition, excellent Esg performance can send a positive signal to the market; the better the Esg performance, the higher the recognition of the external market. This can attract more investors to participate in stock trading [5], which is conducive to lowering the cost of equity capital and reducing the expected risk of insider trading and surplus management by external investors. This will reduce the cost of corporate financing and thus will promote the company's long-term sound growth and development. Based on the above analysis, the first hypothesis is proposed.

**H1:** *Firms' Esg performance positively affects firm growth, and this effect has a lagged effect.*

According to the theory of enterprise innovation, enterprise innovation can improve the competitiveness of an enterprise's business, reduce its operating costs, and improve its productivity and product quality [21]. In terms of environmental protection, enterprise innovation can improve the productivity and efficiency of enterprises, improve resource utilization, reduce the use of resources, and enhance the protection of the environment. At the same time, enterprise innovation can also reduce the carbon emissions of enterprises, thus better realizing the environmental objectives of Esg performance. In terms of social responsibility, enterprise innovation can develop safer, more reliable, and more efficient products, can meet consumer demand for recycled products, and can improve consumer trust and word-of-mouth evaluation of enterprises [22]. Meanwhile, corporate innovation can also strengthen product quality supervision, safeguard employee welfare, etc. In terms of corporate governance, corporate innovation can improve the efficiency of corporate management, realize data-based and intelligent management, better supervise the internal risks of the enterprise [23], and follow a transparent governance mechanism.

The positive impact of innovation on enterprises is multi-faceted; it can play a role in society, the humanities, and the value of various fields. Enterprise innovation is not limited to technology; it can be born in all of the aspects of the enterprise process, from the initial research and development of the product to the goodwill brought about by enterprise innovation to attract more relevant innovative talents and elements [24,25] full of elemental resources to make it possible for enterprises to achieve more substantial innovation. Enterprise innovation creates a cycle of enterprise value and resources. In the production of products, in addition to the innovation of production equipment, enterprise innovation can also be reflected in the transformation and upgrading of the value creation mode and business process, so as to avoid unnecessary production links or costs [26]. In the internal management of the entire enterprise process, enterprise innovation can build a sustainable organizational culture, and managers may have a higher willingness to take risks and assume more social responsibility.

Therefore, corporate innovation can further optimize the Esg performance of enterprises and improve their social responsibility, environmental protection ability, and corporate governance effectiveness and further enhance their development. Based on the above analysis, the second hypothesis is proposed.

**H2:** *Corporate innovation strengthens the positive relationship between Esg performance and firm growth.*

Corporate financing constraints are affected by their long-term business conditions. Companies with good Esg performance tend to demonstrate good, sustainable business development models and philosophies. In the short term, taking on more social responsibility and environmental obligations may increase a company's financing costs [27,28], but in the long term, it is more likely that the company will be recognized by the industry and by the market, which in turn will increase its long-term valuation [29,30].

According to signaling theory, investors in financial markets can make investment decisions by obtaining information about the Esg performance of different firms [31]. If a firm's Esg performance is excellent, it will be able to access more financing resources because investors tend to support firms with better environmental, social, and governance practices. Conversely, if a firm's Esg performance is poor, it will face stricter financing constraints because investors may view it as an unsustainable investment; this could jeopardize the firm's growth [32]. In addition, banks and other financial institutions may also be more willing to lend to firms with good Esg performance because they are more likely to deliver higher returns in the future [33]. Conversely, financial institutions may be more cautious about providing financing to a business with poor Esg performance [34,35], as they may be concerned about the potential risks that environmental, social, and governance factors may pose in the future.

Therefore, if a firm's Esg performance is excellent, it will be able to obtain more financing resources, thus supporting its development, and conversely, it may face stricter financing constraints, thus limiting its development. Based on the above analysis, the third hypothesis is proposed.

**H3:** *A firm's good Esg performance can alleviate the firm's financing constraints and can improve the efficiency of financing, thus promoting the growth of the firm.*

## 3. Research Design

### 3.1. Sample Selection and Data Sources

The sample data for this paper were selected from companies listed on China's Shanghai and Shenzhen exchanges from 2010 to 2020, and the final sample interval was determined to be from 2010 to 2020, from the perspective of data availability. According to the needs of the study, ST and PT companies and companies with missing data on the main variables in the sample period were excluded, and finally, 18,992 samples were obtained. The data were obtained from the CSMAR database and the Wind database, and Stata statistical software version 18 was utilized to shrink the extremes of all the continuous variables on both sides of the 1% quartile to exclude their influence on the results. Meanwhile, in order to verify the accuracy of the database, this paper compares the downloaded data with a sample of company annual reports to ensure the accuracy of the data.

### 3.2. Definition of Variables

#### 3.2.1. Explained Variables

Development means improving the quality and the efficiency of economic development, and thus, the value-added rate of the economy has been widely adopted by academics as an indicator of the level of enterprise development [5,10]. The driving force of development is enterprise innovation, and enterprise innovation ultimately brings about the improvement of enterprise operational efficiency, which is reflected in the improvement of the economic value-added rate in financial indicators. Shareholders, as owners of enterprises, value the input–output ratio, and the rate of economic value added (Reva) can reflect the ratio of capital investment and value created by enterprises. Therefore, it is reasonable to use the rate of economic value added (Reva) to measure the level of

development of enterprises. Most of the existing literature on related issues has adopted Reva as an indicator of enterprise development.

$$\text{Reva} = (\text{Economic Value Added (EVA)}/\text{Total Capital}) \times 100\% \tag{1}$$

### 3.2.2. Explanatory Variables

CSI is one of the three major index companies in China, and it was also one of the early launchers of the Esg indices in China. Its evaluation is based on publicly disclosed data such as the annual reports, periodic reports, and interim announcements of listed companies; the social responsibility and sustainable development reports of listed companies; announcements on the websites of regulatory agencies; news and public opinion data, etc. The CSI Esg rating methodology is constructed on the basis of the rating and scoring models of the international mainstream organizations, incorporating the special national conditions and the actual situations of China. The CSI Esg rating methodology is an exclusive Esg evaluation framework constructed on the basis of the scoring model of mainstream international organizations, and it integrates the special national conditions and the actual situation of China. Therefore, the core explanatory variables of Esg performance in this paper are constructed based on the CSI Esg rating system.

### 3.2.3. Control Variables

| 98 | AAA | AA | A | BBB | BB | B | CCC | CC | C |
|---|---|---|---|---|---|---|---|---|---|
| Corresponding converted score | 9 | 8 | 7 | 6 | 5 | 4 | 3 | 2 | 1 |

The model also controls for the following factors, with reference to existing studies [3,5,10]: mainly firm size (the natural logarithm of total assets), firm growth (the growth rate of main business revenue), gearing Lev (the total liabilities/total assets), operating cash flow Cf (the net cash flow from operating activities over total assets at the beginning of the period), and the nature of the firm's property rights. SOE (when the beneficial controller has state-owned attributes, the variable takes a value of 1, otherwise 0), firm survival Est (the number of years the firm has survived since its inception), and equity concentration Top1 (the percentage of shares held by the first largest shareholder). Refer to Table 1 for specific variable definitions.

**Table 1.** List of variable definitions.

| Variable Name | | Variable Symbol | Description of Variables |
|---|---|---|---|
| Explanatory variable | Economic value-added rate | Reva | Economic growth/total capital |
| Explanatory variable | Esg performance | Esg | CSI Esg Assignment Result |
| Moderator variable | Enterprise Innovation | Rd | Logarithm of firms' R&D investment |
| Intermediary variable | Financing constraints | Sa | Calculated by the multivariate construction index method |
| | Enterprise size | Size | Logarithm of total assets |
| | leverage | Lev | gearing |
| | growth | Gr | Revenue growth rate |
| | Operating cash flow | Cf | Operating cash flow over total assets at beginning of period |
| Control variable | Nature of property rights | SOE | The variable takes the value of 1 for state-owned attributes and 0 otherwise. |
| | Survival period | Est | Number of years since the establishment of the enterprise |
| | Shareholding concentration | Top1 | Shareholding of the largest shareholder as a percentage of total share capital |

### 3.3. Model Setting

This paper focuses on the impact of Esg performance on corporate development. In order to test Hypothesis 1, it is necessary to put the Esg performance of enterprises into



the regression model as an explanatory variable and to put enterprise development as an explanatory variable into the regression model. A regression analysis is conducted to determine whether Esg performance can promote the development of enterprises. Therefore, the model is constructed as follows:

$$\text{Reva} = \beta_0 + \beta_1 \, \text{Esg}_{it} + \text{Controls} + {}_{it}\sum \text{Year} + \sum \text{Industry} + \epsilon_{it} \qquad (2)$$

The model is used to study the impact of Esg performance on firm development. Where Reva denotes the level of enterprise development, Esg denotes the Esg performance of enterprises, i denotes A-share listed companies, and t denotes the year. $\beta_1$ is significantly positive if the Esg performance of enterprises has a significant contribution to the level of enterprise development.

In order to explore the interaction between Esg performance and firm innovation, as well as to more accurately describe the way in which firm development is affected by explanatory and moderating variables, this paper adds C_Esg×Rd, the interaction term between Esg performance and firm innovation after centering, to this model. Therefore, the model is constructed as follows:

$$\text{Reva} = \beta_0 + \beta_1 \, \text{C\_Esg}_{it} + \beta_2 \, \text{C\_Rd\_Esg}_{it} + \beta_3 \, \text{C\_Rd}_{it} + \text{Controls} + {}_{it}\sum \text{Year} + \sum \text{Industry} + \epsilon_{it} \quad (3)$$

The model is used to study the moderating effect of corporate innovation on corporate Esg performance and corporate development levels, where i denotes the listed A-share companies, t denotes the year, Reva denotes the level of enterprise development, C_Esg denotes the Esg performance of enterprises after centralization, C_Rd denotes the level of enterprise innovation after centralization, and C_Esg×Rd denotes the interaction term between the Esg performance of enterprises and the Rd of enterprise innovation after centralization. If firm innovation strengthens the positive impact of Esg performance on firm development, the coefficient $\beta_1$ is significantly positive and $\beta_2$ is also significantly positive.

In order to investigate the mediating effect of Esg performance on the impact of corporate development, the model is constructed as follows:

$$\text{Reva} = \beta_1 \, \text{Esg} + \epsilon_1 \qquad (4)$$

$$\text{Sa} = \beta_2 \, \text{Esg} + \epsilon_2 \qquad (5)$$

$$\text{Reva} = \beta_2 \, \text{Esg} + \beta_3 \, \text{Esg} + \epsilon_3 \qquad (6)$$

The model is used to investigate the way in which firms' Esg performance affects the level of firm development through the financing constraint Sa. In this case, Equation (4) tests the total effect of Esg performance on firm development Reva, Equation (5) tests the role of the independent variable Esg performance on the mediating effect financing constraint Sa, and Equation (6) tries to test the effect of the mediating variable Sa on Reva and the effect of Esg on Reva after controlling for the mediating effect, where $\varepsilon_1$, $\varepsilon_2$, and $\varepsilon_3$ are independent of each other with white noise. The above model three is regressed sequentially on Equations (4)–(6) by the stepwise test method, and the Sobel test is given for the joint coefficients. If the coefficients $\beta_1$, $\beta_2$, $\beta_3$ are significant, it indicates the presence of a mediation effect.

## 4. Empirical Results and Analysis

### 4.1. Descriptive Statistics and Correlation Analysis

Table 2 presents the statistics of the model variables, including the statistics of the sample, mean, standard deviation, and the maximum and minimum values. Table 2 shows that the average economic value-added rate of enterprise development is 0.01; the great and small values are 0.23 and −0.48, respectively, indicating that there is a

large gap in the development level of China's listed enterprises. The mean value of Esg performance reaches 6.48, the overall performance is good, and the mean value is located in the middle of the range of ratings from one to nine, but there is still a certain gap from the high-level Esg performance which is consistent with the actual situation that the Esg development in China's development is in the preliminary stage; the greater the company's corporate innovation Rd, indicating that the company's R&D investment in the proportion of operating income, of which the smallest is 13.27, the largest is 21.93, the largest and smallest value of the extreme difference is large, reflecting the existence of the sample enterprises with a significant degree of corporate innovation; financing constraints take the absolute value of the smallest of −4.39, the largest of −3.08, and the difference between them is −4.39 and −3.08. The difference between the two is very small, reflecting the sample enterprise financing constraints of the smaller differences.

**Table 2.** Descriptive statistics of main variables.

| Variables | Sample Size | Average Value | (Statistics) Standard Deviation | Minimum Value | Upper Quartile | Maximum Values |
|---|---|---|---|---|---|---|
| Reva | 18,992 | 0.01 | 0.07 | −0.48 | 0.01 | 0.23 |
| Esg | 18,992 | 6.48 | 1.10 | 3 | 6 | 9 |
| Size | 18,992 | 22.23 | 1.18 | 20.04 | 22.06 | 26.21 |
| Sa | 18,992 | −3.79 | 0.22 | −4.39 | −3.79 | −3.08 |
| Rd | 18,992 | 17.90 | 1.44 | 13.27 | 17.93 | 21.93 |
| Lev | 18,992 | 0.41 | 0.19 | 0.06 | 0.41 | 0.90 |
| Cf | 18,992 | 0.05 | 0.06 | −0.14 | 0.05 | 0.23 |
| Gro | 18,992 | 0.28 | 0.54 | −0.65 | 0.14 | 4.27 |
| Top1 | 18,992 | 33.73 | 14.10 | 8.54 | 31.56 | 74.30 |
| SOE | 18,992 | 0.34 | 0.47 | 0 | 0 | 1 |
| Est | 18,992 | 17.37 | 5.35 | 5 | 17 | 32 |

The variables were analyzed for correlation results, as shown in Table 3. The coefficient between the level of enterprise development and Esg performance is significantly positive, which indicates that Esg performance has a positive impact on the level of enterprise development. At the same time, the correlation coefficient between the Esg indicators and the control variables in the model is small, which indicates that the problem of multicollinearity is not serious.

**Table 3.** Table of correlation coefficients for each variable.

| | Reva | Esg | Size | Cf | Lev | Gro | Top1 | Est | Sa | Rd |
|---|---|---|---|---|---|---|---|---|---|---|
| Reva | 1.000 | | | | | | | | | |
| Esg | 0.152 *** | 1.000 | | | | | | | | |
| Size | 0.127 *** | 0.312 *** | 1.000 | | | | | | | |
| Cf | 0.391 *** | 0.099 *** | 0.071 *** | 1.000 | | | | | | |
| Lev | −0.106 *** | 0.067 *** | 0.504 *** | −0.142 *** | 1.000 | | | | | |
| Gro | −0.026 *** | −0.001 | −0.079 *** | −0.128 *** | −0.020 *** | 1.000 | | | | |
| Top1 | 0.118 *** | 0.152 *** | 0.171 *** | 0.096 *** | 0.068 *** | −0.046 *** | 1.000 | | | |
| Est | −0.007 | 0.049 *** | 0.201 *** | 0.028 *** | 0.116 *** | −0.036 *** | −0.089 *** | 1.000 | | |
| Sa | −0.002 | 0.006 | −0.106 *** | −0.033 *** | −0.091 *** | 0.026 *** | 0.116 *** | −0.516 *** | 1.000 | |
| Rd | 0.192 *** | 0.181 *** | 0.502 *** | 0.118 *** | 0.160 *** | −0.013 * | 0.021 *** | 0.064 *** | −0.056 *** | 1.000 |

$* \, p < 0.1$, $*** \, p < 0.01$.

## 4.2. Regression Analysis

First, using the least squares method and gradually adding control variables and year and industry fixed effects, the benchmark model is regressed, and it is initially verified that Esg performance positively promotes the development of enterprises. The regression estimation results are shown in Table 4. The regression results are shown in column (1). The regression coefficient of Esg performance is positive and is significant at the 1% level,

still reflecting the positive impact of Esg performance on the high level of enterprise development. Next, on the basis of this regression, the control variables of corporate finance, operation, and governance are added to the regression, and the regression results are shown in column (2). The independent variable of corporate Esg performance is also significant at the 1% level. The paper conducts a Hausman test, using a two-way fixed-effects model for individual and year effects for the regression, and the results are shown in column (3), with a coefficient of 0.0032 for corporate Esg performance, which is still significant at the 1% level of significance. Statistically, Esg performance shows a significant positive correlation with the level of corporate development, verifying research Hypothesis 1.

**Table 4.** Esg performance on firm development regression results.

|  | **(1)** | **(2)** | **(3)** |
|---|---|---|---|
|  | **Reva** | **Reva** | **Reva** |
| Esg | 0.0076 *** | 0.0042 *** | 0.0032 *** |
|  | (0.0004) | (0.0005) | (0.0007) |
| SOE |  | −0.0154 *** | −0.0163 *** |
|  |  | (0.0011) | (0.0038) |
| Size |  | 0.0052 *** | 0.0244 *** |
|  |  | (0.0006) | (0.0014) |
| Lev |  | −0.0342 *** | −0.0992 *** |
|  |  | (0.0029) | (0.0054) |
| Cf |  | 0.3891 *** | 0.2364 *** |
|  |  | (0.0083) | (0.0096) |
| Gro |  | 0.0039 *** | 0.0058 *** |
|  |  | (0.0010) | (0.0011) |
| Top1 |  | 0.0004 *** | 0.0006 *** |
|  |  | (0.0000) | (0.0001) |
| Est |  | −0.0000 | −0.0030 *** |
|  |  | (0.0001) | (0.0003) |
| _Cons |  | −0.2452 *** | −0.4910 *** |
|  |  | (0.0100) | (0.0285) |
| Time effect | yes | yes | yes |
| Individual effect | yes | yes | yes |
| $R^2$ | 0.0171 |  | 0.1357 |
| $N$ | 18,992 | 18,992 | 18,992 |
| Hausman |  | 441.48 *** |  |

Standard errors in parentheses. *** $p < 0.01$.

### 4.3. Impact Path Regression Analysis

In order to test the moderating role of the moderating variable corporate innovation between the explanatory variables and the explained variables, this section conducts a regression analysis of the moderating effect of corporate innovation based on Model II, and the regression results are shown in Table 5.

**Table 5.** Regression statistics of the moderating effect of firm innovation.

|  | **Reva** |
|---|---|
| c_Esg | 0.0049 *** |
|  | (0.0006) |
| c_Rd | 0.0046 *** |
|  | (0.0005) |
| c_Esg_Rd | 0.0012 ** |
|  | (0.0005) |
| SOE | −0.0157 *** |
|  | (0.0018) |
| Size | 0.0079 *** |

**Table 5.** *Cont.*

|  | Reva |
| --- | --- |
|  | (0.0008) |
| Lev | −0.0647 *** |
|  | (0.0040) |
| Cf | 0.3057*** |
|  | (0.0088) |
| Gro | 0.0041 *** |
|  | (0.0010) |
| Top1 | 0.0006 *** |
|  | (0.0000) |
| _Cons | −0.1678 *** |
|  | (0.0172) |
| Time effect | yes |
| Individual effect | yes |
| $R^2$ | 0.1995 |
| $N$ | 18992 |

Standard errors in parentheses. ** $p < 0.05$, *** $p < 0.01$.

The results of the test in the table above show that, by observing the significant level of the regression variables, and after adding the moderator variable, firm innovation, and the interaction term between it and the explanatory variable, Esg performance, to the model (4-2), the regression coefficient of firms' Esg performance is 0.0049, which is significant at the 1% level, indicating that the positive and facilitating effect of Esg performance on the development of the firms maintains a significant relationship. A positive contribution maintains a significant relationship. The regression coefficient of the interaction term between Esg performance and firm innovation is 0.0012, which is significant at the 5% level. The positive coefficient of this interaction term indicates that corporate innovation has a reinforcing effect on the positive role of Esg performance in enterprise development; that is to say, the higher the level of corporate innovation of an enterprise, the more the Esg performance of the enterprise plays a facilitating role in promoting the development of the enterprise, controlling for the same conditions as the other variables. This result verifies research Hypothesis 2.

In this paper, regression analysis was conducted to test the mediating effect of financing constraints based on Model III, as shown in Table 6.

**Table 6.** Statistical results of the regression of the mediating effect of financing constraints.

|  | (1) | (2) | (3) |
| --- | --- | --- | --- |
|  | Reva | Sa | Reva |
| Esg | 0.0032 *** | −0.0099 *** | 0.0029 *** |
|  | (0.0007) | (0.0010) | (0.0006) |
| Sa |  |  | −0.0520 *** |
|  |  |  | (0.0057) |
| Controls | yes | yes | yes |
| Year | yes | yes | yes |
| Industry | yes | yes | yes |
| $R^2$ | 0.1357 | 0.0241 | 0.1433 |
| N | 16,791 | 16,791 | 16,791 |
|  | Sobel-Goodman Mediation test | | |
|  | Coef | Z | $p > |Z|$ |
| Sobel | −0.00024001 | −8.795 | 0.0001 |

Standard errors in parentheses. *** $p < 0.01$.

The results are shown in the table above: column (1) is the basic regression results, and column (2) indicates the role of Esg performance on financing constraints. The regression results show that the correlation between Esg performance and the mediating variable

is negative and significant at 1% level, which indicates that Esg performance and the financing constraints are negatively correlated, and the good Esg performance of the enterprise can alleviate the financing constraints of the enterprise; as shown in column (3), the Esg performance indicator and the financing constraint indicator Sa are added into the model at the same time, and the mediating effect played by financing constraints is initially verified. The Esg performance indicator and the financing constraint indicator Sa are added to the model at the same time; the coefficient of the mediating variable Sa is significantly negative at the 1% level, and the mediating effect played by the financing constraint is initially verified. At the same time, this paper adopts the Sobel test to further analyze the mediation effect. As shown in the figure, the Z value is −8.795, which verifies the existence of the mediation effect, and the result is significant at the 1% level. In summary, the hypothesis is verified. This result indicates that the good Esg performance of enterprises alleviates their financing constraints, improves financing efficiency, and promotes their development. Thus, it strengthens the support for Hypothesis 3.

*4.4. Robustness Tests and Treatment of Endogeneity Problems*

4.4.1. Considering the Lag in Esg Performance

In this paper, the explanatory variable, firm Esg performance, is lagged one period to overcome the endogeneity problem caused by two-way causality. The results are shown in Table 7, where the regression coefficients of the lagged one-period and lagged two-period explanatory variables LEsg are significantly positive at the 10% level in both cases, indicating that the main regression results are robust.

**Table 7.** Robustness test results.

| Variables | (1) Current Period | (2) One Period Behind | (3) Phase II Lag | (4) Three-Phase Lag (in Technology) |
|---|---|---|---|---|
| Esg | 0.0032 *** | 0.000162 * | 0.000103 * | 0.000929 |
| | (−0.0007) | (−0.03) | (−0.66) | −0.25 |
| SOE | −0.0163 *** | −0.0093 *** | −0.0071 *** | −0.0046 *** |
| | (−0.0038) | (−0.22) | (−0.66) | (−0.25) |
| Size | 0.0244 *** | 0.122 *** | 0.148 *** | 0.132 *** |
| | (−0.0014) | (−11.42) | (−10.86) | (−16.51) |
| Lev | −0.0992 *** | −0.704 *** | −0.857 *** | −0.572 *** |
| | (−0.0054) | (−21.01) | (−21.09) | (−24.81) |
| Cf | 0.2364 *** | 0.243 *** | 0.258 ** | 0.283 *** |
| | (−0.0096) | (−3.69) | (−3.24) | (−6.19) |
| Gro | 0.0058 *** | 0.000114 | 0.0000606 | 0.000015 |
| | (−0.0011) | (−0.38) | (−0.18) | (−0.09) |
| Top1 | 0.0006 *** | 0.00065 | 0.000131 | 0.000605 |
| | (−0.0001) | (−0.93) | (−0.15) | (−1.15) |
| Est | −0.0030 *** | −0.0157 *** | −0.0191 *** | −0.0164 *** |
| | (−0.0003) | (−7.95) | (−7.60) | (−11.08) |
| _Cons | −0.4910 *** | −2.189 *** | −2.598 *** | −2.472 *** |
| | (−0.0285) | (−10.03) | (−9.23) | (−14.76) |
| Time effect | yes | yes | yes | yes |
| Individual effect | yes | yes | yes | yes |
| $R^2$ | 0.1357 | 0.05 | 0.11 | 0.06 |
| N | 18992 | 14493 | 11903 | 9600 |

Standard errors in parentheses. * $p < 0.10$, ** $p < 0.05$, *** $p < 0.01$.

4.4.2. Replacement of Core Variables

In order to determine whether the empirical results of Reva, the proxy variable selected earlier in this paper to measure the level of the development of firms, are robust, Progress I selects other proxy variables to substitute for Reva. Tobin's Q (the ratio of firms' stock

market capitalization to firms' replacement cost) is used to measure the development level of firms. The regression results show that the coefficients between the explanatory variables, Esg performance and TBQ, are positive and significant at the 1% level, the signs of the coefficients of the other control variables are consistent with the baseline regression results in the previous section, and they are all significant at a high level, thus further testing the robustness of the model (Table 8).

**Table 8.** Results of replacement of variables.

|  | TbQ |
|---|---|
| Esg | 0.104 *** |
|  | (7.42) |
| SOE | −0.159 *** |
|  | (−3.75) |
| Size | 0.0244 *** |
|  | (0.0014) |
| Lev | −0.015 *** |
|  | (−12.87) |
| Cf | 0.030 *** |
|  | (11.07) |
| Gro | 0.001 *** |
|  | (7.05) |
| Top1 | 0.009 *** |
|  | (7.28) |
| Est | −0.0120 *** |
|  | (0.0001) |
| _Cons | 12.056 *** |
|  | (29.77) |
| Time effect | yes |
| Individual effect | yes |
| $R^2$ | 0.401 |
| N | 18,863 |

Standard errors in parentheses. *** $p < 0.01$.

### 4.4.3. Tool Variables

This paper does not exclude the effect of endogeneity, so the instrumental variable method and two-stage least squares (2SLS) are used for further testing. The average of the Esg scores of all firms in the province where the sample firm is located except for that firm (Esg-IV) is selected as the instrumental variable. Since enterprises located in the same province have the same policy conditions, resource endowment, and industrial cluster effect, the Esg performance of this enterprise has a certain connection with the average of the Esg ratings of enterprises in this province, but the Esg performance of other enterprises has no direct connection with the level of development of a single enterprise, so it meets the criteria for the selection of instrumental variables. The regression results are shown in Table 9.

**Table 9.** Instrumental variable test results.

|  | First Stage | Second Stage |
|---|---|---|
|  | **Esg** | **Reva** |
| Esg-IV | 0.775 *** |  |
|  | (0.0317) |  |
| Esg |  | 0.024 *** |
|  |  | (0.0033) |
| Size | 0.282 *** | 0.034 *** |
|  | (0.0083) | (0.0035) |
| Lev | −0.836 *** | −0.366 *** |

**Table 9.** *Cont.*

|  | First Stage | Second Stage |
|---|---|---|
|  | Esg | Reva |
|  | (0.0498) | (0.0190) |
| Cf | 1.065 *** | 0.083 * |
|  | (0.1348) | (0.0462) |
| Gro | 0.033 ** | 0.000 |
|  | (0.0147) | (0.0000) |
| Top1 | 0.002 *** | 0.001 *** |
|  | (0.0005) | (0.0002) |
| Est | 0.003 * | −0.001 ** |
|  | (0.0016) | (0.0006) |
| SOE | 0.338 *** | −0.000 |
|  | (0.0193) | (0.0086) |
| _Cons | −4.677 *** | 12.056 *** |
|  | (0.2428) | (0.0116) |
| Time effect | yes | yes |
| Individual effect | yes | yes |
| $R^2$ | 0.1981 | 0.2107 |
| N | 16440 | 16440 |

Standard errors in parentheses. * $p < 0.10$, ** $p < 0.05$, *** $p < 0.01$.

## 5. Further Analysis

### 5.1. Classification by Nature of Shareholding

The impact of corporate Esg performance on corporate development may vary in degree, depending on the nature of the enterprise's ownership. State-owned enterprises (SOEs) are the representatives of the government in the economy and, therefore, their Esg performance is more strictly regulated by the government. In contrast, the Esg performance of non-state-owned enterprises may be subject to less government regulation. As an important part of the economy and society, SOEs need to assume more social responsibility. Non-SOEs may tend to be more spontaneous in their behavior towards society, the environment, and governance. Therefore, while Esg is an important consideration for all firms, the degree of influence may differ between non-SOEs and SOEs. Based on these factors, this paper argues that the development of SOEs and non-SOEs may be affected by Esg performance to a slightly different extent (Table 10).

**Table 10.** Further analysis results—by nature of shareholding.

|  | State-Owned Holding | Non-State Holding |
|---|---|---|
| Esg | 0.00474 * | 0.0215 *** |
|  | (−2.30) | (−6.31) |
| Size | 0.0158 *** | 0.0127 *** |
|  | (−9.01 | (−3.96) |
| Lev | −0.124 *** | −0.00287 |
|  | (−13.45) | (−1.25) |
| Cf | 0.713 *** | 0.315 *** |
|  | (−27.19) | (−8.00) |
| Gro | 0.000000468 | 0.000091 |
|  | (−0.13) | (−1.45) |
| Top1 | 0.0000652 | 0.00126 *** |
|  | (−0.42) | (−5.22) |
| Est | −0.000868 * | −0.000634 |
|  | (−2.16) | (−1.07) |
| _Cons | −0.355 *** | −0.455 *** |
|  | (−10.02) | (−6.69) |

**Table 10.** *Cont.*

|  | State-Owned Holding | Non-State Holding |
|---|---|---|
| N | 6457 | 12535 |
| $R^2$ | 0.11 | 0.01 |
| Adj. $R^2$ | 0.11 | 0.01 |

Standard errors in parentheses. * $p < 0.10$, *** $p < 0.01$.

*5.2. Classification by Nature of Industry*

Similarly, the impact of an enterprise's Esg performance on its development will vary depending on the industry in which it operates. Compared with traditional industries, high-tech enterprises pay more attention to environmental protection, employee welfare, social welfare, etc. Meanwhile, high-tech enterprises need to pay more attention to their own corporate innovation, development, and other factors. This industry characteristic may lead to a more significant impact of their Esg performance on enterprise value. Based on these factors, this paper argues that the development of high-tech industries and non-high-tech industries may be affected by Esg performance to a slightly different extent (Table 11).

**Table 11.** Heterogeneity analysis results—Classification by industry nature.

|  | High-Tech Industries | Non-High-Tech Industries |
|---|---|---|
| Esg | 0.0246 *** | 0.0026 |
|  | (0.0033) | (0.0038) |
| Size | 0.0342 *** | 0.0683 *** |
|  | (0.0035) | (0.0064) |
| Lev | −0.3668 *** | −0.5105 *** |
|  | (0.0190) | (0.0251) |
| Cf | 0.0837 * | 0.2938 *** |
|  | (0.0462) | (0.0510) |
| Gro | 0.0000 | 0.0000 |
|  | (0.0000) | (0.0001) |
| Top1 | 0.0012 *** | 0.0007 |
|  | (0.0002) | (0.0004) |
| SOE | −0.0009 | −0.0081 |
|  | (0.0086) | (0.0158) |
| Est | −0.0014 ** | −0.0044 *** |
|  | (0.0006) | (0.0011) |
| _cons | −0.8102 *** | −1.1506 *** |
|  | (0.0714) | (0.1138) |
| N | 3998 | 14994 |

Standard errors in parentheses. * $p < 0.10$, ** $p < 0.05$, *** $p < 0.01$.

The results of the analysis show that the Esg performance of enterprises with different property rights has different degrees of influence on the explanatory variables and that the development level of non-state-controlled enterprises is influenced by Esg performance to a greater extent compared to the development of state-owned enterprises, which is influenced by Esg performance to a lesser extent; the development of enterprises in different industries is influenced by Esg performance to different extents; the development of high-tech enterprises is more significantly affected by Esg performance.

## 6. Discussion

This paper presents an empirical study on the relationship between Esg performance, financing constraints, innovation, and firm development. It is found that Esg performance can significantly promote enterprise development. The road test finds that Esg performance can promote enterprise development by alleviating financing constraints and then promote enterprise development, while innovation can strengthen the promotion effect of Esg

performance on enterprise development. After robustness tests, such as instrumental variables and lagged effects, the research conclusions still hold.

From the perspective of the capital market, firms with higher Esg performance tend to be more willing to disclose high-quality information; this can not only inhibit the short-sightedness of managers but can also reduce the intrinsic motivation of corporate surplus management, inhibit disclosure violations, and improve the financial performance and market value of the enterprise [36,37]. This is somewhat similar to the findings of this paper.

From the perspective of internal governance, it is basically consistent with the findings of Guangyou et al. [10] that enterprises with good Esg performance can attract more investment opportunities to alleviate financing constraints and can bring more financial support to enterprises, which in turn promotes enterprise development. From the perspective of resource allocation, Wang et al. [17] believe that technological innovation can change the existing resource allocation model and can promote resource utilization; in addition, Feng et al. [38] believe that collaborative innovation can help enterprises break the industry boundaries and resource constraints, triggering the knowledge spillover effect and facilitating the formation of innovation networks, which in turn promotes the sustainable development of enterprises. This is somewhat similar to the conclusion of this paper that innovation strengthens Esg performance on enterprise development.

## 7. Conclusions

With the increasing global recognition of Esg performance, Esg performance has begun to become one of the criteria for measuring the dynamic balance of ecological and economic structures, and more and more enterprises have begun to actively practice Esg concepts. Against this background, this study empirically investigates the impact of Esg performance on corporate development using listed Chinese A-share companies from 2010 to 2020 as the research sample. The study finds that, first, Esg performance has a significant positive contribution to corporate development, and the municipal structure remains robust through data lags, replacement variables, and instrumental variables. Second, path analysis shows that Esg performance can promote firm development by reducing financing constraints; innovation can strengthen this promotion effect. Third, further analysis reveals that Esg performance has a more significant role in promoting the development of firms in non-state-controlled and high-tech industries.

Based on the above findings, this paper puts forward the following suggestions:

Firstly, enterprises should strengthen the construction of Esg concepts to promote corporate Esg performance. From the perspective of enterprises, maintaining a dynamic balance between ecological and economic structures has become a general trend. Enterprises should accelerate the implementation of Esg concepts in all business processes and value creation processes, practice social responsibility, regulate and constrain corporate behavior, and then achieve sustainable corporate development. In addition, path analysis found that innovation can enhance the positive impact of Esg performance on enterprise development. Therefore, enterprises should increase investment in innovation and improve the quality of corporate innovation to accelerate the efficiency of corporate development.

Secondly, different enterprises and industries have significant heterogeneity in Esg performance, due to differences in size, geography, nature of property rights, level of competition, and other factors, which is particularly significant for non-state-controlled and high-tech industries. To address this situation, the relevant departments can formulate special Esg subsidy policies and audit strategies, take non-state-controlled and high-tech industry enterprises as a pilot, and gradually expand to the whole industry to accelerate the Esg process for the sustainability of the enterprise and benign development of empowerment.

**Author Contributions:** Conceptualization, L.L. and Q.H.; literature review, L.L. and Y.S.; methodology, D.M. and M.W.; Software, L.L. and Y.S.; validation, D.M. and M.W.; formal analysis, D.M., L.L. and Q.H.; investigation, Y.S. and Q.H.; resources, M.W.; data curation, Q.H.; writing—original

draft preparation, L.L. and Q.H.; writing—review and editing, D.M. and L.L.; supervision, D.M.; All authors have read and agreed to the published version of the manuscript.

**Funding:** This research received no external funding.

**Institutional Review Board Statement:** Not applicable.

**Informed Consent Statement:** Not applicable.

**Data Availability Statement:** The raw data supporting the conclusions of this article will be made available by the authors, without undue reservation.

**Conflicts of Interest:** The authors declare no conflict of interest.

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
