# Peer review of "Corporate Sustainability: The Impact of Environmental, Social, and Governance Performance on Corporate Development and Innovation"

_sustainability, doi:10.3390/su151914086_

Round 1

Reviewer 1 Report

From my point of view, the analysis of the manuscript was conducted (almost entirely) according to the model presented by Hua Tang in the article "The Effect of ESG Performance on Corporate Innovation in China: The Mediating Role of Financial Constraints and Agency Cost" (Sustainability 2022,14(7), 3769; https://doi.org/10.3390/su14073769).

At this moment, I consider that the current form cannot be published in the journal Sustainability because it is not an original contribution.

The authors can consider a reorganization of the manuscript, in an original version, and then it can be considered for publication.

Thank you!

Reviewer 2 Report

The article leaves a pleasant impression regarding its scientificity and the robustness of the results obtained on the basis of the applied methods. However, certain points give grounds for separate recommendations in terms of the possibility of its improvement.

The abstract does not indicate the purpose of the conducted research, the main tasks that need to be solved in order to achieve an idicated goal. .

It is not correct to ask questions in the abstract.

It would not hurt to specify the purpose of the research also in the introduction.

The conclusions are too general in nature, they contain already quite well-known information, which is obviously connected with a rather imprecise expression of the purpose of the research.

It is appropriate to link the conclusions with the tasks and hypotheses and show their verification and achievement

Author Response

请参阅附件。

Reviewer 3 Report

Please consider the attached file.

Round 2

Reviewer 1 Report

Dear authors,

First of all, thank you for your answer!

However, your answer seems to be formulated to my general remark and not to specific comments.As I noted in the pdf attached to the previous report, I mostly referred to the structure of the work.Thus, I am sorry that you did not respond to any of the recommendations formulated.

I agree with the publication of the work considering that certain changes have been made.

I wish you good luck!